# Experimental Validation of High Precision Web Handling for a Two-Actuator-Based Roll-to-Roll System

**DOI:** 10.3390/s22082917

**Published:** 2022-04-11

**Authors:** Jaeyoung Kim, Kyungrok Kim, Hyunchang Kim, Pyoungwon Park, Seonghyun Lee, Taikmin Lee, Dongwoo Kang

**Affiliations:** Department of Flexible and Printed Electronics, Korea Institute of Machinery and Materials, Daejeon 34103, Korea; kr88kim@kimm.re.kr (K.K.); hckim0128@kimm.re.kr (H.K.); pwpark81@kimm.re.kr (P.P.); shlee79@kimm.re.kr (S.L.); taikmin@kimm.re.kr (T.L.); dwkang@kimm.re.kr (D.K.)

**Keywords:** roll-to-roll (R2R), cascade control, velocity control, tension control, radius estimation algorithm, air floating roller

## Abstract

In this paper, experimental validation of high precision web handling for a two-actuator-based roll-to-roll (R2R) system is presented. To achieve this, the tension control loop is utilized to regulate the tension in the unwinder module, and the velocity loop is utilized to regulate the web speed in the rewinder module owing to the limitation of the number of actuators. Moreover, the radius estimation algorithm is applied to achieve an accurate web speed and the control sequence of the web handling in the longitudinal axis is developed to manipulate the web handling for convenience. Having these, the tension control performances are validated within ±0.79, ±1.32 and ±1.58 percent tension tracking error and 1.6, 1.53 and 1.33 percent web speed error at the speeds of 0.1 m/s, 0.2 m/s, and 0.3 m/s, respectively. The tension control performance is verified within ±0.3 N tracking error in the changes of the reference tension profile at 0.1 m/s web speed. Lastly, the air floating roller is used to minimize the friction terms and the inertia of the idle roller in the tension zone so that tension control performance can be better achieved during web transportation.

## 1. Introduction

The web processing in a roll-to-roll (R2R) system introduces an attractive manufacturing process that has certain advantages over mass production—fast manufacturing, a low cost, convenient, and reliable method compared to other manufacturing processes [1]. The R2R system handles a long flexible material such as plastic films, flexible panels, papers, etc. so that it can manufacture large-scale transistors, light-emitting diode (LED) displays, organic light-emitting diodes (OLED) displays, and other electric devices [2,3,4,5,6,7,8].

The types of web processing in the R2R system include coating, lamination, heating, slitting, etc. [9,10,11,12,13]. The major process of the R2R system is that the web should be properly transported on rollers; the raw material of the web, which is wrapped on the first roller, needs to be unwound, transported through the R2R system for web processing, and it needs to rewind on the last roller. Therefore, this system generally consists of winders (rewinder and unwinder), idle rollers, and driven rollers. Winders are used to rewind or unwind the web during the continuous manufacturing process. These winders are a critical and dominant process of web handling in the R2R system. Idle rollers are used to support the web and achieve the desired web path through processing machinery. Driven rollers are used to control the web speed so that the speed difference of the rollers can be minimized.

To achieve a high web processing performance in the R2R system, web handling is a critical factor in the web transporting behavior. Specifically, the web longitudinal and lateral dynamics are the main concerns from the perspective of the control area. In the longitudinal dynamics of web transportation in the R2R system, the desired velocity and tension should be maintained during web transportation. Therefore, control designs of the web speed and tension are one of the important objectives to achieve mass production with high quality in the R2R system.

In the past, numerous studies have been conducted to design the control of velocity and tension of the web. H. Koc et al. [14] designed a linear parameter varying (LPV) control strategy with smooth scheduling for various desired references and an $H_{\infty }$ control strategy with varying gains to reduce the coupling between tension and velocity of the web and change of roll radius. PR. Pagilla et al. [15] designed a decentralized control strategy utilizing a proportional–integral (PI) control and linear quadratic control for the velocity and tension of the web. Moreover, various control strategies (ex. adaptive feedforward control, disturbance rejection control, resonance filter, etc.) were utilized to minimize the presence of the non-ideal roller effect, external disturbance from the mechanical components, and change of roller’s radius, etc. [16,17,18,19,20,21,22,23]. However, to control the web speed during transportation, most works are relevant to design the control algorithm to reduce the effect of change of the roller’s radius which is assumed as an external disturbance. Therefore, it is necessary to handle directly the change of the roller’s radius so that the precise web speed can be maintained during web transportation under the limited number of actuators.

In our research, a two-actuator-based R2R system is developed to operate web transportation in the longitudinal direction. Compared to the conventional R2R system, there is no driven roller or feeder so only winders should operate to regulate the tension and web speed simultaneously. Therefore, the rewinder module is responsible for the velocity control and the unwinder module is responsible for the tension control, respectively, in our system. For tension control, the cascade control strategy is utilized in which the velocity control loop is located in the inner loop and the tension control loop is located in the outer loop; the tension control loop is utilized as Proportional–Integral–Derivative (PID) control. For the velocity control, Proportional–Integral (PI) control with a notch filter is utilized. Owing to the change in the radius for each winder, the radius estimation algorithm is developed and applied to the control sequence so that the tension control with a controlled web speed can be simultaneously achieved for convenience. With the control strategies and the experimental setup, the tension tracking performances of the web transportation in the longitudinal direction are experimentally validated under several scenarios. Having these control strategies enables easier and more convenient manipulation of the R2R system with a good tension tracking control performance under the controlled web speed using only two actuators.

This paper is organized as follows. The system structures of the R2R system and its major components are presented in Section 2. The dynamic models of the web span and each winder are presented in Section 3. The control design, estimation radius algorithm, and control sequence are described in Section 4. Section 5 presents an experimental hardware setup for the R2R system, and the experimental validations of the tension tracking control performance under several scenarios are presented in Section 6. Section 7 presents the conclusion and future recommendations.

## 2. System Structure

The system structure of the R2R system is presented as shown in Figure 1.

This system consists of rewinder (diameter: 75 mm, total shaft length: 382 mm), unwinder (diameter: 75 mm, total shaft length: 382 mm), 8-idle rollers (diameter: 60 mm, width: 196 mm), 2-loadcells, and one incremental rotary encoder. For rewinding and unwinding parts, a steel-based lug with a mechanical shaft is used to hold the core tight so that it can minimize the slip between the core and shaft; the interaction of the core and the shaft are assumed as one rigid body. The core of winders (diameter: 96 mm, length: 208 mm) is made up of aluminum so that the lug from the shaft can hold the core without any deformation of its shape. The PET film (thickness: 0.18 mm, width: 150 mm) is used to operate in web handling and the total diameter of the rewinder including the full film is about 166 mm. The loadcell, which is located in the rewinder module as shown in Figure 1, is the sensor for providing the measurement to the tension feedback control loop and another loadcell, which is located in the unwinder module, is the sensor for monitoring. An incremental rotary encoder is used to monitor the web speed so that it can verify the radius estimation algorithm under the desired web speed. The total length of the web span in our system is about 2260 mm.

## 3. Dynamic Models

### 3.1. Web Dynamic Model

In this section, dynamic models of major components in the R2R system are presented. To describe the mathematical model of the web tension model, as shown in Figure 2, several assumptions are established as follows: (1) The length of the contact region between the web material and a roller is negligible compared to the length of the free web span between the rollers. (2) The thickness of the web is very small compared with the radius of rollers over which the web is wrapped. (3) There is no slippage between the web material and the rollers. (4) There is no mass transfer between the web material and the environment. (5) The strain (ε) in the web is much smaller than unity. (6) The strain is uniform within the web span. (7) The web cross-section area (*A*) in the unstretched state does not vary along with the web. (8) The density (ρ) and the modulus of elasticity (*E*) of the web in the unstretched state are constant over the cross-section. (9) The web is perfectly elastic. (10) The web properties do not change with temperature or humidity.

According to the law of conservation of mass, the equation can be derived as:(1)$$\frac{d}{dt}\left(\int_{x_{i}}^{x_{i+1}}\rho (x,t)A(x,t)dx\right)=\rho_{i}A_{i}v_{i}-\rho_{i+1}A_{i+1}v_{i+1}$$
where $x_{i}$ and $x_{i+1}$ are the inlet and outlet position of the span and, *v* is the velocity of the web. It is noted that the $i_{th+1}$ web span is located between the $i_{th}$ and $i_{th+1}$. The subscript “u” on the variable represents the unstretched state of the variable. With the relationship between the stretched length and unstretched length in the web and the mass equation, these two equations can be combined as:(2)$$\frac{\rho (x,t)A(x,t)}{\rho {\left(x,t\right)}_{u}A{\left(x,t\right)}_{u}}=\frac{dx_{u}}{dx}=\frac{dx_{u}}{\left(1+\varepsilon_{i+1}\right)dx_{u}}=\frac{1}{\left(1+\varepsilon_{i+1}\right)}$$

Based on the assumption (6) and (7), Equations (1) and (2) combined can be expressed with the only function of time and it is simplified as:(3)$$\frac{d}{dt}\left(\int_{x_{i}}^{x_{i+1}}\frac{1}{1+\varepsilon_{i+1}\left(t\right)}dx\right)=\frac{v_{i}\left(t\right)}{1+\varepsilon_{i}\left(t\right)}-\frac{v_{i+1}\left(t\right)}{1+\varepsilon_{i+1}\left(t\right)}=L\frac{d}{dt}\left(\frac{1}{1+\varepsilon_{i+1}\left(t\right)}\right)$$
where *L* is the web span length between rollers. With assumption (5), $\frac{1}{1+\varepsilon (t)}$ is closed to 1−ε(t) and the above equation can be derived as:(4)$$L\frac{d}{dt}\left(\varepsilon_{i+1}\left(t\right)\right)=-v_{i}\left(t\right)+v_{i+1}\left(t\right)+\varepsilon_{i}\left(t\right)v_{i}\left(t\right)-\varepsilon_{i+1}\left(t\right)v_{i+1}\left(t\right)$$

With a small perturbation approach ($\Delta \varepsilon \left(t\right)=\varepsilon \left(t\right)-\varepsilon_{0},\Delta v\left(t\right)=v\left(t\right)-v_{0}$) and initial nominal operation ($0=-v_{i,0}+v_{i+1,0}+\varepsilon_{i,0}v_{i,0}-\varepsilon_{i+1,0}v_{i+1,0}$) with Δε(t)·Δv(t)<<1, the above equation is linearized as follows.
(5)$$L\frac{d}{dt}\left(\Delta \varepsilon_{i+1}\left(t\right)\right)=v_{i,0}\Delta \varepsilon_{i}\left(t\right)-v_{i+1,0}\Delta \varepsilon_{i+1}\left(t\right)-\Delta v_{i}\left(t\right)+\Delta v_{i+1}\left(t\right)$$

With a Hooke’s law ($\Delta T_{i+1}\left(t\right)=AE\Delta \varepsilon_{i+1}\left(t\right)$), the finalized dynamic equation for the web tension model can be derived as:(6)$$\frac{d}{dt}\left(\Delta T_{i+1}\left(t\right)\right)=\frac{v_{i,0}\Delta T_{i}\left(t\right)}{L}-\frac{v_{i+1,0}\Delta T_{i+1}\left(t\right)}{L}+AE\frac{\Delta v_{i+1}\left(t\right)-\Delta v_{i}\left(t\right)}{L}$$

With $v_{i,0}$, $v_{i+1,0}$ are assumed as the velocity in the steady transportation ($v_{s}$) and $\Delta T_{i}\left(t\right)$ is closed to zero; the plant model ($P_{tension}$) from the input and output can be expressed as:(7)$$P_{tension}=\frac{T(s)}{v(s)}=\frac{AE}{Ls+v_{s}}$$
where T(s) and v(s) are the Laplace transform of $\Delta T_{i+1}\left(t\right)$ and $\Delta v_{i+1}\left(t\right)-\Delta v_{i}\left(t\right)$, respectively. The detailed derivations of the web tension model are described in [23,24].

### 3.2. Dynamic Models of Major Components

The dynamic models of the major components in the R2R system are presented in this section. Based on the configurations shown in Figure 3, the dynamic model of the unwinder is derived as:(8)$$\frac{d}{dt}\left(J_{u}\left(t\right)v_{u}\left(t\right)\right)=R_{u}{\left(t\right)}^{2}T_{i}\left(t\right)-bv_{u}\left(t\right)-R_{u}\left(t\right)\tau_{u}\left(t\right)$$
where $J_{u}\left(t\right)$ is the inertia of the unwinder, $v_{u}\left(t\right)$ is the velocity of the unwinder ($=w_{u}\left(t\right)\cdot R_{u}\left(t\right)$), *b* is the viscous friction coefficient, $R_{u}\left(t\right)$ is the radius of the unwinder, $T_{i}\left(t\right)$ is the tension generated by torque from the motor to the unwinder, and $\tau_{u}\left(t\right)$ is the torque transmitted from the motor to the unwinder.

The inertia of the unwinder can be expressed as $J_{u}\left(t\right)=J_{0}+J_{w}\left(t\right)$ ($J_{0}$: the inertia of the motor, and $J_{w}\left(t\right)$: the inertia of the web). $J_{w}$ can be expressed in terms of changing the radius as:(9)$$J_{w}\left(t\right)=\frac{m_{w}}{2}\left(R_{u}{\left(t\right)}^{2}+R_{c}^{2}\right)=\frac{\pi }{2}\rho_{w}\left(t\right)W\left(R_{u}{\left(t\right)}^{2}-R_{c}^{2}\right)\left(R_{u}{\left(t\right)}^{2}+R_{c}^{2}\right)$$
where, $m_{w}$ is the mass of the web, $R_{c}$ is the radius of core roller, $\rho_{w}\left(t\right)$ is the density of the web and, *W* is the width of the web. With a differentiation of the inertia of the web material ($\dot{J_{w}}\left(t\right)=2\pi \rho_{w}\left(t\right)WR_{u}{\left(t\right)}^{3}\dot{R_{u}}\left(t\right)$) and a differentiation of the radius ($\dot{R_{u}}\left(t\right)=\frac{-t_{span}v_{u}\left(t\right)}{2\pi R_{u}\left(t\right)}$, $t_{span}$: thickness of the web), the dynamic equation of the unwinder can be re-expressed as:(10)$$J_{u}\left(t\right){\dot{v}}_{u}\left(t\right)=R_{u}{\left(t\right)}^{2}T_{i}\left(t\right)-bv_{u}\left(t\right)-R_{u}\left(t\right)\tau_{u}\left(t\right)+\rho_{w}\left(t\right)Wt_{span}R_{u}{\left(t\right)}^{2}v_{u}{\left(t\right)}^{2}$$

For the dynamic model of the rewinder, it can be derived from the consideration of increasing the radius of the roller and acting in the opposite direction of tension as:(11)$$J_{r}\left(t\right){\dot{v}}_{r}\left(t\right)=-R_{r}{\left(t\right)}^{2}T_{i+n}\left(t\right)+R_{r}\left(t\right)\tau_{r}\left(t\right)-bv_{r}\left(t\right)-\rho_{w}\left(t\right)Wt_{span}R_{r}{\left(t\right)}^{2}v_{r}{\left(t\right)}^{2}$$
where $J_{r}\left(t\right)$ is the inertia of the rewinder, $v_{r}\left(t\right)$ is the velocity of the rewinder, $R_{r}\left(t\right)$ is the radius of the rewinder, $T_{i+n}\left(t\right)$ is the tension generated by torque from the motor to the rewinder, and $\tau_{r}$ is the torque transmitted from the motor to the rewinder. For the dynamic model of the idle roller using the torque balance, it can be expressed as:(12)$$J_{ir}{\dot{v}}_{ir}\left(t\right)=R_{ir}^{2}\left(T_{i+n}\left(t\right)-T_{i}\left(t\right)\right)-\tau_{f,ir}\left(t\right)R_{ir}$$
where $J_{ir}$ is the inertia of the idle roller, $v_{ir}\left(t\right)$ is the velocity of the idle roller, $R_{ir}$ is the radius of the idle roller, and, $\tau_{f,ir}\left(t\right)$ is the friction torque owing to the rolling friction, viscous friction, and surface friction. The detailed dynamic equations of the major components in the R2R system are derived in [16,25].

## 4. Control Design

The objective of the web handling in the longitudinal axis for the R2R system is to transport the web under a controlled web speed and tension. Therefore, one of the critical factors in our system is to design the controllers so that the velocity and tension variations should be simultaneously minimized during web transportation. As shown in Figure 4, in the unwinder module, the cascade control strategy is utilized; the velocity control loop is located in the inner loop and the tension control loop is located in the outer loop. In the cascade control strategy, the control bandwidth of the inner loop is at least 5 times faster than that of the outer loop to achieve a faster system response. For the rewinder module, velocity control is utilized to maintain the constant web speed.

Changing the radius of each winder directly affects the change of the inertia of the roller so that the system responses of each winder with a full film and the empty film are different. As shown in Figure 5, the bode plot of the plant and closed-loop transfer function (CLTF) of the roller with the full film and empty film are presented using the frequency response function (FRF) of the experimental data based on the excitation swept sinusoidal signals. From the bode plot of the plant, the first anti-resonance of the roller with the full-film and empty film occurs at 142 Hz and 448 Hz, respectively, and the resonance peaks in the high frequencies occur at around 1350 Hz for both cases. To eliminate these peaks, the notch filter is designed so that the magnitude of the resonance peaks can decrease by 25 dB for both cases. For the velocity control algorithm, Proportional–Integral (PI) control is utilized to reduce the variation between the reference and the measurement from the encoder embedded in the motor. Since the inertia of the roll with the full film is larger than that of the roll with empty film, higher control gains are needed to minimize the variation of the velocity in the roll with full film. However, the inertia of the rewinder and unwinder can be exchangeable so that the control gains are optimized to be the same for both winders. From the CLTF bode plots, the control bandwidth of the roller with the full film is approximately 45 Hz and that of the empty roller is approximately 247 Hz, respectively. That is, the control bandwidth of the tension control should be at least 5 times less than the control bandwidth of the roller with full film. For the tension control, it is utilized as proportional–integral–derivation (PID) control to minimize the variation between the reference tension and measurement from the loadcell. To adjust the control gains, the Ziegler–Nichols tuning method is used [26]. Based on the web dynamic model, the plant from the velocity and the tension is a first-order transfer function so the Ziegler–Nichols tuning method can be applied in the tension control tuning. To apply this, the ultimate gain ($k_{u}$) is found to be marginally stable for the closed-loop transfer function of the outer loop. Then, the oscillation period ($T_{u}$) is used to set the gains. The control gains of the tension loop are obtained as $K_{p}=0.6k_{u}$, $K_{i}=2K_{p}/T_{u}$ and $K_{d}=K_{p}T_{u}/8$. Therefore, the entire control input from the feedback control ($u_{FB}\left(t\right)$) and feedforward compensator ($u_{FF}\left(t\right)$) of the tension loop can be expressed as:(13)$$u_{total}\left(t\right)=u_{FB}\left(t\right)+u_{FF}\left(t\right)=K_{p}e\left(t\right)+K_{i}\int e(t)dt+K_{d}\frac{de(t)}{dt}+\frac{V_{ref}}{R_{u}\left(t\right)}$$
where, $K_{p}$, $K_{i}$, and $K_{d}$ are control parameters of proportional gain, integration gain, and derivative gain. $V_{ref}/R_{u}\left(t\right)$ is the feedforward compensator computed by the change of radius and the web speed. The derivation of the feedforward compensator is presented in the radius estimation algorithm. According to the bode plot of the CLTF for the tension control loop in Figure 6, the control bandwidth is about 9 Hz so the specification of the cascade control strategy is achieved, of which the control bandwidth of the outer loop should be at least five times less than that of the inner loop.

To estimate the change of the winder’s radius, a simplified radius estimation algorithm is applied to the rewinder and the unwinder under the assumption of constant strain during web transportation. Based on the reference web speed ($V_{web}=V_{ref}\left(t\right)=R\left(t\right)w\left(t\right)$), the length of the web span is derived as:(14)$$\begin{array}{c} L\left(t\right)=\int_{0}^{t}V_{web}\left(t\right)dt=\int_{0}^{t}R\left(t\right)w\left(t\right)dt \end{array}$$
where w(t) is the angular velocity of the winder and R(t) is the radius of the winder, respectively. With the length of the web span, the updated radius of each winder can be derived as:(15)$$\begin{array}{ccc} R_{u}\left(t\right) & =\sqrt{-\frac{t_{span}\int_{0}^{t}R_{u}\left(t\right)w_{u}\left(t\right)dt}{\pi }+R_{o}^{2}},\;R_{r}\left(t\right) & =\sqrt{\frac{t_{span}\int_{0}^{t}R_{r}\left(t\right)w_{r}\left(t\right)dt}{\pi }+R_{c}^{2}} \end{array}$$
where $R_{o}$ is the outer radius of the unwinder and $R_{c}$ is the core radius of each winder, respectively. With the reference web speed and the updated radius of the rewinder, the reference angular velocity ($w_{ref}\left(t\right)$) can be applied to the velocity control loop. For the angular velocity ($w_{o})$ computed by the reference web speed and the updated radius of the unwinder, it is utilized as the feedforward compensator in the tension control loop. Since the radius of the winder is inevitably changing during the web transportation, the tension control output can be changed if only the PID feedback controller is utilized; the web is limited to transport to another roller under good tension control performance. Therefore, using the feedforward control based on the change in the roller’s radius, the tension control effort can be reduced and the total length of the web can be more transported to another roller under good tension control performance. If the current radius of the roller is measured incorrectly, the tension control performance can be degraded during the long period of web transportation. Based on the control designs and the updated radius of the winder, the control sequence is developed to operate the web handling in the longitudinal axis well so that better tension control performance can be achieved as well as more convenient manipulation of the R2R system. The control sequence of a high accuracy tension control in the R2R system is presented as shown in Figure 7 and a detailed explanation of each diagram is presented below.
(1)The motor drive (velocity control loop) and the upper-level controller (tension control loop) are activated.(2)The reference web speed and the tension are applied to the zero under the control algorithm (impose control).(3)The ramp signal of the reference tension is applied to the rewinder so that the web can be smoothly forced to enact the tension without the undesired film expansion.(4)The ramp signal of the reference speed is applied to each winder so that the angular velocity (computed by the web speed and the updated radius of the roller) is smoothly generated as the applied input.(5)The two-actuator-based R2R system can operate under the good control performance of the tension and web speed.

## 5. Experimental Setup

An experimental setup for a high accuracy web transportation of the two-actuator-based R2R system is presented in Figure 8. The real-time interface (RTI) on dSPACE was used to obtain the measurement and to control the web tension. The dSPACE controller board (DS1007) is suitable for high-precision control owing to the flexibility of developing the control algorithms using MATLAB/Simulink. Moreover, data acquisition in real-time can be performed. Velocity control was utilized from the motor drive (AKD-B00606 from Kollmorgen), which generates the angular velocity of the winder based on the reference angular velocity from the DS1007 R&D controller board. For the winder, the direct-drive rotary (DDR) motor (C042A-13-3305 from Kollmorgen) was directly mounted to the shaft to be driven, thereby eliminating the external disturbances from the mechanical transmission elements. For the tension measurement, the loadcell with its amplifier (TSA Loadcell Amplifier of MAGPOWR) was used to obtain the tension measurement in the rewinder and unwinder module and it provided the tension control loop to achieve a good control performance of the feedback controller. To verify the web speed in our system, the incremental rotary encoder (E50S8-8000-3-T-5 from Autonics) was installed in the rewinder module and it provided the upper-level controller for monitoring. The sampling frequency of the signals in the DS1007 R&D controller board was about 10 kHz.

## 6. Experimental Results

With the control strategies and experimental setup for the two-actuator-based R2R system, the experimental validations were tested under several scenarios. The control performance of the tension loop was validated at different speeds of the web transportation during 50 s of the recording time. The reference tension was 38 N and the web speed was set up at 0.1 m/s, 0.2 m/s, and 0.3 m/s, respectively.

For the tension and velocity control performance at 0.1 m/s of the web speed, the tracking error of the tension control loop is about ±0.3 N which represents a ±0.79 percentage tracking error of the desired tension value (38 N) as shown in Figure 9a,b. The total control output of the tension loop including the feedforward compensator and the feedback controller increases from about 1.25 V to 1.3 V during web transportation, as shown in Figure 9c. For the control performance of the web speed by applying the radius estimation algorithm, it is necessary to measure the web speed from the incremental rotary encoder in the full web transportation (from the empty to the full film). In this scenario, the total length of the web during web transportation used is about 65.7 m and other conditions are the same as the previous setup.

The control performance of the web speed at 0.1 m/s is shown in Figure 10a. This web speed was measured by the incremental rotary encoder with the radius of the idle roller. The average web speed in the first 2 s is about 0.100 m/s and that in the last 2 s which is about 0.1016 m/s. The web speed error is increased by 1.6 percent compared to the desired web speed in full web transportation. From Figure 10b, it can be seen that the initial radius of the rewinder is about 48 mm and is increased to 78.224 mm by applying the radius estimation algorithm. Compared to the measured radius of the rewinder at the end of the experiment (78.37 mm), the error between the estimated radius of the rewinder and the measured radius of the rewinder is about 0.186 percent. Therefore, it is shown that the good control performance of the web speed is verified using the proposed radius estimation algorithm in the two-actuator-based R2R system. The tension and velocity control performance at different levels of the web speed is presented in Table 1.

In the manufacturing process of the R2R system, the changes in the reference tension might be applied during the web transportation such as starting point of the web handling in the longitudinal axis in the processing zone, tension tapering, etc. Therefore, it is important to verify the tension control performance by applying the changes in the reference tension during web transportation. The changes in the reference tension profile are presented in Figure 11a. The reference tension starts at 48 N and is decreased proportionally to 28 N over 20 s. After the tension is maintained at 28 N for 10 s, it is increased proportionally to 38 N and the tension value is maintained at rest. For the web speed, it is set up as 0.1 m/s. Comparing the reference tension profile and the measurement as shown in Figure 11b, the tracking error is about ±0.3 N which is a similar level to the tension control performance in Figure 9a. Therefore, it proves that good tension control performance is verified within the proportional changes in the reference tension during web transportation.

Owing to the inertia and friction terms of the idle rollers in the tension zone, the tension control performance might be degraded so that the quality of the product manufactured in the R2R system can be poor. Moreover, the tension fluctuation occurred due to the eccentricity of the idle rollers. In this scenario, the static and dynamic tension control performance is verified using the air floating roller in the tension zone to minimize the undesired disturbance of the idle rollers. The air floating rollers (75 mm Air Turn of Newway) are used to levitate the web in the tension zone as shown in Figure 12. The wrap angle of the air floating roller is about 110 degrees and the input pressure is set up as 0.4 MPa among the viable pressure range. To verify the dynamic performance of the web using an air floating roller, the confocal sensor (sensor probe with IFC2421/2422 of Micro-Epsilon) is used to measure the air gap between the web and roller’s surface. The reference tension is 38 N and the web speed is set up at 0.1 m/s, respectively. From Figure 13a,b, the peak-to-peak magnitude of the tension output using the air floating roller in the tension control loop is about 0.62 N, which is reduced by about 0.1 N compared to that of tension output using the idle roller only. Moreover, the tension difference between the rewinder and unwinder module is closed to zero (0.085 N) using the air floating roller; it is verified that the static friction term can be minimized using the air floating roller in the tension zone. For verifying the dynamic performance of web transportation which is measured by the confocal sensor as shown in Figure 13c, the air gap between the roller’s surface and the web is about 148.2 μm to the 181 μm and, the tension variation is about 32.8 μm.

## 7. Conclusions

In this paper, experimental validation of a high accuracy web handling in the longitudinal axis for a two-actuator-based roll-to-roll (R2R) system is presented. To achieve this, the tension control loop uses a cascaded control strategy in the unwinder module; the velocity control loop is located in the inner loop (PI control) and the tension control loop is located in an outer loop (PID control). The velocity control loop is utilized in the rewinder module to maintain the web speed owing to the limitation of the number of actuators. To consider the changes in the radius for the winders, the radius estimation algorithm is applied to the rewinder and the unwinder to achieve the precise web speed and to generate the feedforward compensator in the tension control loop. Moreover, the control sequence of the web handling in the longitudinal axis is developed to manipulate the R2R system for convenience. With the control methods and experimental setup, the tension control performances of the two-actuator-based R2R system are validated under ±0.79, ±1.32 and ±1.58 percentage tracking error at the web speed of 0.1 m/s, 0.2 m/s and 0.3 m/s, respectively. For the control performance of the web speed, it is verified using the incremental rotary encoder during full web transportation and the web speed errors are 1.6, 1.53 and 1.33 percent at the web speed of 0.1 m/s, 0.2 m/s and 0.3 m/s, respectively. Moreover, the radius estimation error is also verified within 1 percentage compared to the measured roller radius for all cases. It is also verified that the tension control performance is well maintained within the proportional changes in the reference tension profile. Lastly, to reduce the inertia and friction terms of the idle roller, an air floating roller is used in the tension zone. The experimental results show that static friction terms of the web handling are reduced by about 0.1 N and tension variation between the roller and the web measured by the confocal sensor is about 32.8 micrometers.

The R2R system generally consists of a rewinder, unwinder, and main roller (or in-feeder and out-feeder) so that the constant tension and web speed are at least maintained under the controlled system in the tension zone (processing zone). Therefore, it is our future work to compare the control performance of the web handling including the main roller (or in-feeder and out-feeder) and these in our system with the proposed control strategies. Moreover, with the current radius estimation algorithm, the web speed error can be increased as the radius of the winder wrapped by the web is increased during web transportation. Therefore, it is necessary to develop an additional algorithm that can minimize the increase in web speed error during web transportation. Lastly, as the web speed is increased, the tension control performance worsens owing to the increase in the film vibration, so advanced control strategies are necessary to develop to minimize the undesired disturbances at high speed.

## Figures and Tables

**Figure 1 sensors-22-02917-f001:**
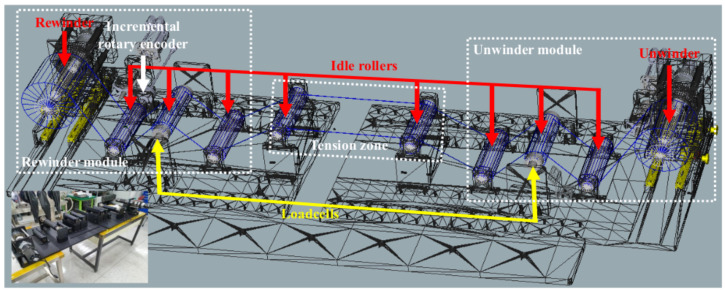
System structure of the two-actuator-based R2R system.

**Figure 2 sensors-22-02917-f002:**
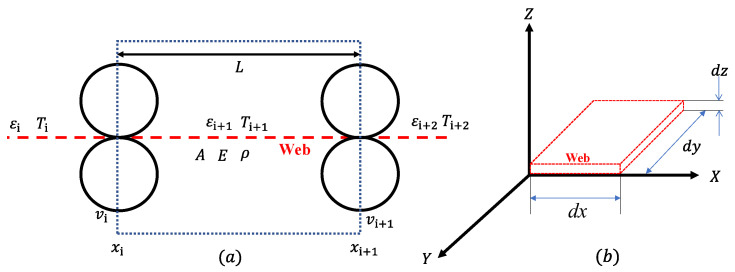
Schematic diagram of the web tension model in the R2R system: (**a**) 2-dimension of the web tension model, and (**b**) 3-dimension of the web tension model.

**Figure 3 sensors-22-02917-f003:**
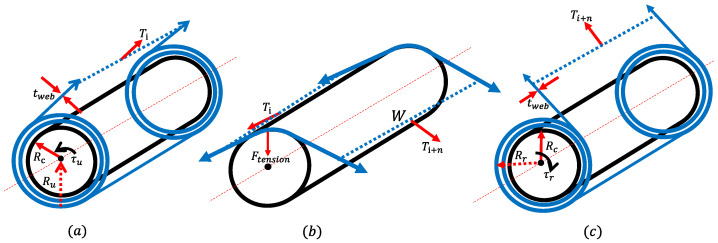
Schematic configuration of major components in the R2R system: (**a**) unwinder, (**b**) idle roller, and (**c**) rewinder.

**Figure 4 sensors-22-02917-f004:**
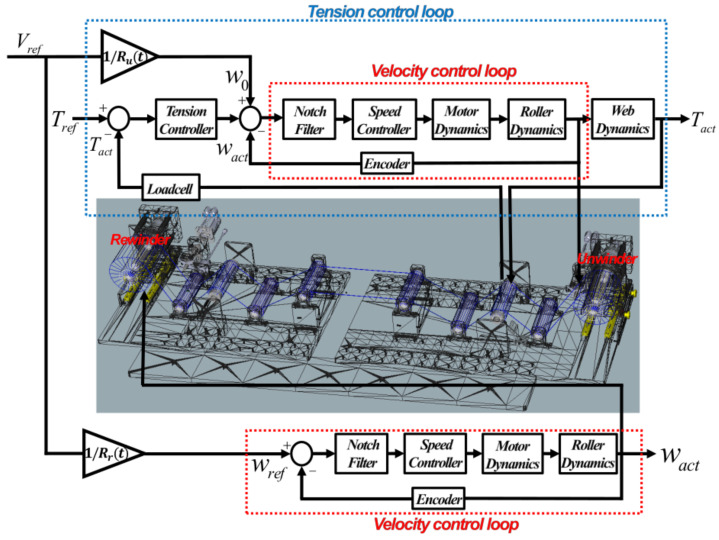
The entire control block diagram in the R2R system.

**Figure 5 sensors-22-02917-f005:**
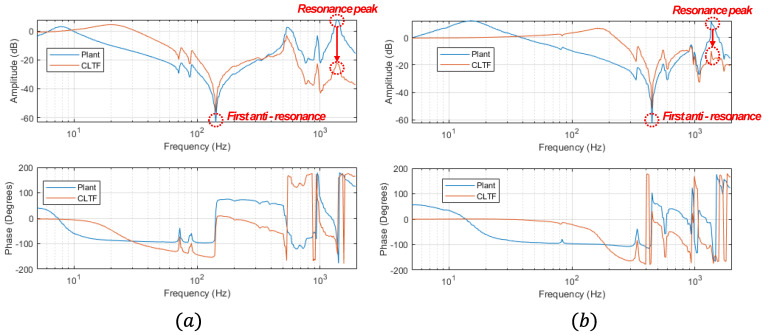
The comparison between the bode plots of each winder: (**a**) the bode plot of the roller with the full film and (**b**) empty film.

**Figure 6 sensors-22-02917-f006:**
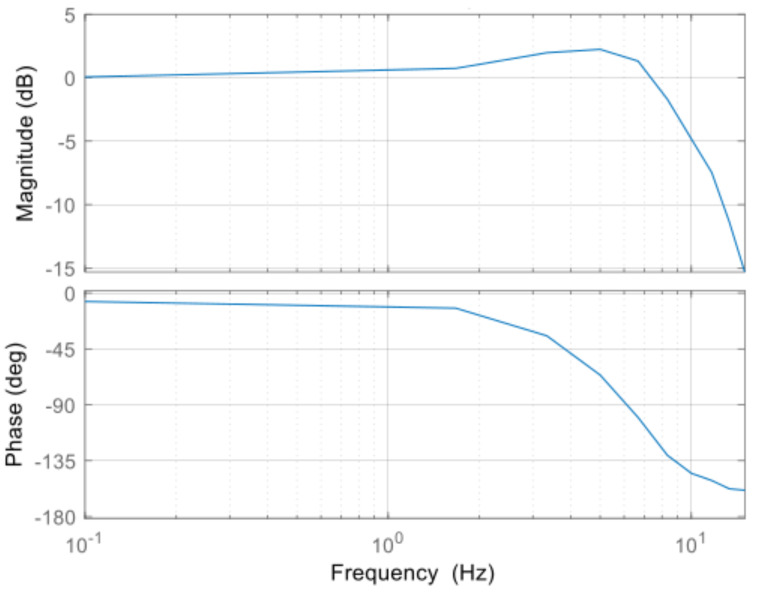
The bode plot of the CLTF for the tension control loop.

**Figure 7 sensors-22-02917-f007:**
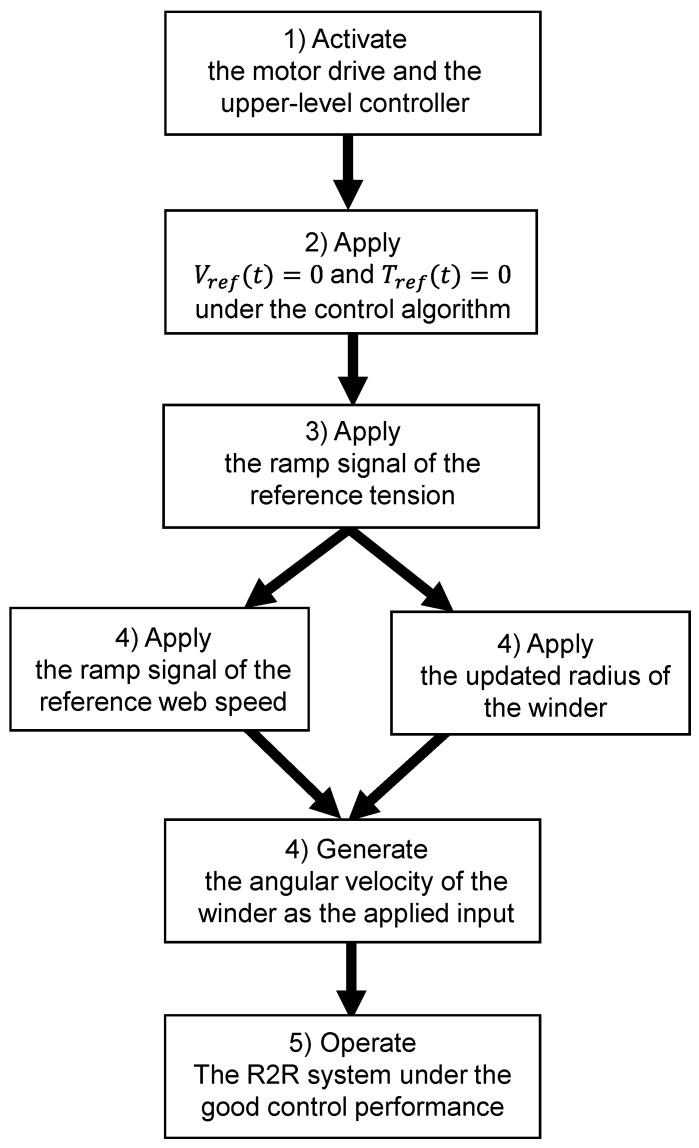
The entire control sequence of web handling in the longitudinal axis for the R2R system.

**Figure 8 sensors-22-02917-f008:**
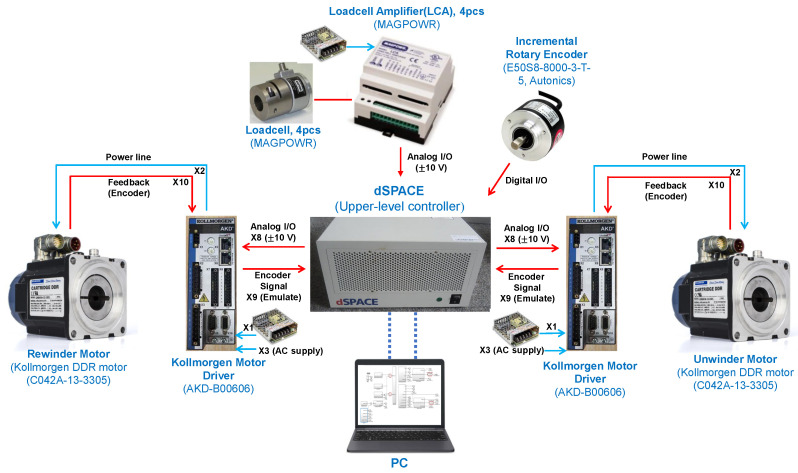
An experimental setup for a high accuracy web transportation of the two-actuator-based R2R system.

**Figure 9 sensors-22-02917-f009:**
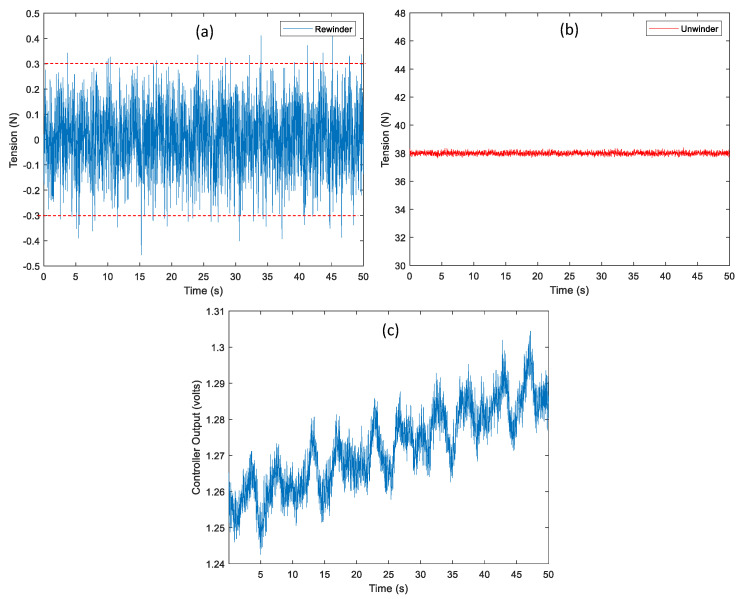
The control performance of the tension at 0.1 m/s of the web speed: (**a**) The tracking error in the tension control loop, (**b**) tension measurement from the loadcell in the unwinder module (tension feedback) and (**c**) control output of the tension control loop.

**Figure 10 sensors-22-02917-f010:**
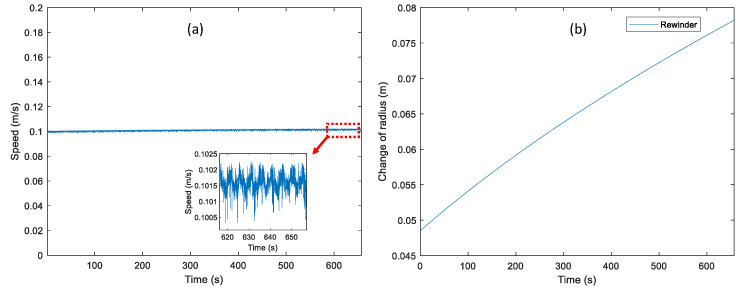
The control performance of the velocity loop is validated at 0.1 m/s of the web speed: (**a**) the web speed measured by the incremental rotary encoder and (**b**) the estimated radius of the rewinder.

**Figure 11 sensors-22-02917-f011:**
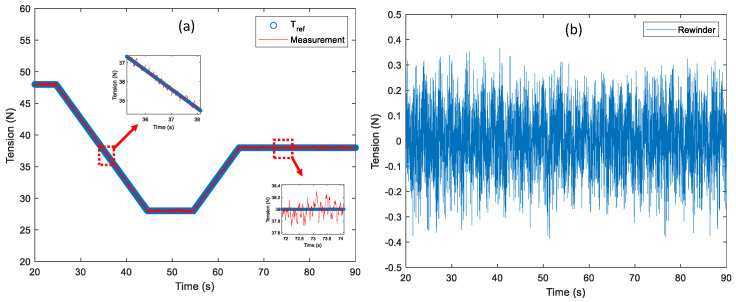
The control performance of the changes in the reference tension is validated at 0.1 m/s of the web speed: (**a**) comparison between the reference tension and measurement and (**b**) tension tracking error.

**Figure 12 sensors-22-02917-f012:**
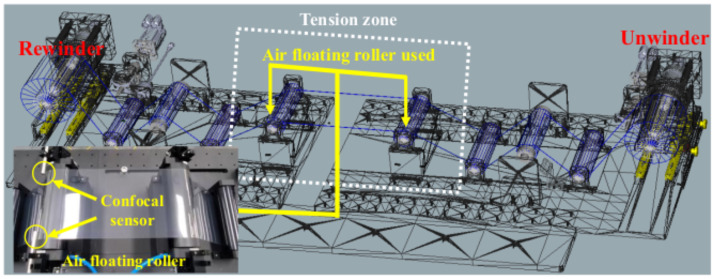
Air floating roller set up in the tension zone.

**Figure 13 sensors-22-02917-f013:**
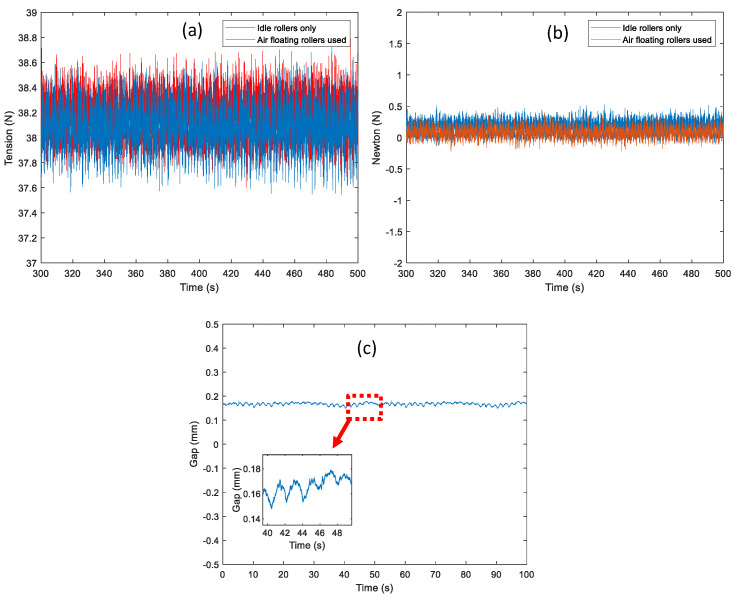
Comparison of the control performance between the idle roller only and the air floating roller used: (**a**) tension measurement of the unwinder module (tension feedback), (**b**) tension difference between the unwinder and rewinder module and (**c**) air gap measurement between the web and roller using the confocal sensor.

**Table 1 sensors-22-02917-t001:** Tension and velocity control performance at different web speeds ($T_{ref}$: 38 N, $V_{ref}$: 0.2 and 0.3 m/s).

Control Performance at 0.2 m/s	Values	Control Performance at 0.3 m/s	Values
Tension tracking error	±1.32%	Tension tracking error	±1.58%
Control output (50 s)	2.55 V to 2.7 V	Control output (50 s)	4 V to 4.4 V
Average web speed (first 2 s)	0.200	Average web speed (first 2 s)	0.300
Average web speed (last 2 s)	0.203	Average web speed (last 2 s)	0.304
Web speed error	1.5%	Web speed error	1.33%
Radius at initial	48 mm	Radius at initial	48 mm
Estimated radius at last	79.19 mm	Estimated radius at last	78.42 mm
Measured radius at last	78.38 mm	Measured radius at last	79.33 mm
Radius estimation error	1.03%	Radius estimation error	1.15%

## Data Availability

Not applicable.

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
