# Peer review of "Experimental Validation of High Precision Web Handling for a Two-Actuator-Based Roll-to-Roll System"

_sensors, 2022, doi:10.3390/s22082917_

Round 1
Reviewer 1 Report
The authors designed a tension control loop to regulate the tensions in the unwinder and rewinder modules. They also experimentally verified the performance of the tension controller that they designed. However, the reviewer cannot find out any advantages of the proposed method considering the previous researches as follows. Especially, only single span that is less complex than the multi-spans introduced in the below researches was considered. The reviewer thinks that the improved point of the study should be clearly emphasized to publish in this journal.
- P. R. Pagilla, R. V. Dwivedula, and N. B. Siraskar, “A Decentralized model reference adaptive Controller for Large-Scale Systems,” IEEE-ASME trans. mech., vol. 12.2, pp. 154-163, 2007
- T. Nishida, T. Sakamoto, N. I. Giannoccaro, “Self-tuning control using adaptative PSO of a web transport system with overlapping decentralized control,” IEEJ Trans. Ind. Appl., vol. 131-D, pp.1442-1450, 2011.
- V. Gassmann, D. Knittel, P. R. Pagilla, and M.-A. Bueno, “Fixed-Order H∞ Tension Control in the Unwinding Section of a Web Handling System Using a Pendulum Dancer,” IEEE trans. Contr. Syst. T., Vol. 20, pp. 173-180, 2012.
- P. R. Raul, S. G. Manyam, P. R. Pagilla, and S. Darbha, “Output regulation of nonlinear systems with application to roll-to-roll manufacturing systems,” IEEE-ASME trans. mech., vol. 20, pp.1089-1098, 2015.
- N. I. Giannoccaro, T. Sakamoto, I. Uchitomi, “A gain scheduling of PI controllers of a multi-span web transport system,” Int. J. Smart Sensing Intell. Syst., vol. 9.3, pp. 1516-1533, 2016.
Reviewer 2 Report
1、For the sake of clarity, I suggest to add a table at the beginning with the main nomenclature used in the paper.
2、In formula 4-7, will the superposition of multiple assumptions affect the correctness of the final model? How to ensure the validity of assumptions in practical application?
3、When the tension reference changes, the controller can maintain good performence in this paper; If the speed reference value changes, does the model work correctly?
4、The estimated value of feedforward compensation includes system parameters. If the parameter measurement is inaccurate, will it affect the control effect?
Reviewer 3 Report
Manuscript entitled “Experimental Validation of a High Precision Web Handling for two Actuators-based Roll-to-Roll System” gives an interesting and systematic research about the development of a two actuators-based roll-to-roll (R2R) system to operate web transportation in the longitudinal direction. Presented results showed that in comparison to the conventional R2R system the winders should be operated in order to regulate the tension and web speed simultaneously; there is no need for the driven roller or feeder.
The topic is original and relevant in the field, because web processing in a R2R system is a part of an attractive manufacturing process mainly used in coating, lamination, heating and slitting procedures and in order to achieve a high processing performance, web handling is a critical factor in the transporting behavior of the material.
The manuscript is well organized, authors used the scientific methods for the conducted analysis and they were adequately described. In the experimental and results part experimental validation of a high accuracy web handling in the longitudinal axis for two actuator-based roll-to-roll (R2R) system is presented. The results are clearly presented and the discussions support presented results. In order to achieve the goal of the research the tension control loop is utilized a cascaded control strategy in the unwinder module; the velocity control loop is located in the inner-loop (PI control) and the tension control loop is located in an outer-loop (PID control). The experimental validations are tested under several scenarios and the authors have proven that by those control strategies it is easier and more convenient the manipulation of the R2R system.
The most of the concluded observations are written in the discussions which I find appropriate for this research. In conclusion the main result is expressed and the direction of the future work are described.
The available published papers in this area report on the various control strategies to control of the velocity and tension of the web, listed in reference part. So, the references used in the research are appropriate.
The only suggestion for the authors is that the Figure 4 should be improved, to make the R2R system more readable to the audience.
The acceptance of the manuscript is suggested with the above mention corrections.
Round 2
Reviewer 1 Report
The paper has improved. And the requested points were correctly addressed.
But I think a little bit more information about
1) the control strategy and control design, and utilized filter are very normal approaches to attenuate tension disturbance of a moving web regarding winder radius change/unknown winder shape. So, what is major benefit through the proposed method?
2) What is the effect of floatation pressure on tension of a moving web? Normally, the total strain of a web in a floating span is function of elastic strain due to speed differences and induced strain by floatation pressure.
